# Understanding the Differing Impacts of On-Call Work for Males and Females: Results from an Online Survey

**DOI:** 10.3390/ijerph16030370

**Published:** 2019-01-29

**Authors:** Bernadette Roberts, Grace E. Vincent, Sally A. Ferguson, Amy C. Reynolds, Sarah M. Jay

**Affiliations:** Appleton Institute, Central Queensland University, Adelaide 5034, Australia; bernie.roberts@live.com.au (B.R.); g.vincent@cqu.edu.au (G.E.V.); sally.ferguson@cqu.edu.au (S.A.F.); a.reynolds@cqu.edu.au (A.C.R.)

**Keywords:** on-call, stand-by, coping, female, male, domestic, work-life balance

## Abstract

On-call work is prevalent worldwide and is associated with adverse outcomes, including disrupted sleep, impaired leisure time, and difficulties in mentally detaching from work. Limited studies specifically explored whether men and women experience on-call differently; therefore, our aim was to investigate whether sex differences exist in terms of both the impacts of and coping strategies to deal with on-call work. On-call workers (*n* = 228) participated in an online survey to investigate how on-call work impacts domestic, non-domestic, and leisure activities, and coping strategies. Pearson chi-squared analyses were used to determine sex differences for each construct of interest. Results indicated that female respondents were more likely to be responsible for running their household, and reported that being on call disturbed leisure, domestic, and non-domestic activities “a lot/very much”. While both males and females adopted engaged coping styles, a greater proportion of males used “problem solving” and a greater proportion of females “talked about their feelings” when managing on-call work. These findings provide valuable insight into how males and females are differentially impacted and cope with on-call work. Further research is required to better understand these impacts, particularly over time, and should include measures such as of quality of life, relationship satisfaction, and physical and mental health outcomes.

## 1. Introduction

On-call operations, which require workers to be available to receive calls and/or attend work if needed [1], afford organizations 24/7 staffing coverage, often overnight or on weekends [2]. Global estimates of on-call work are around 20% or more in employed workforces [3,4]. Furthermore, on-call work is crucial in order to be able to leverage the workforce required for fire and emergency rescue and response [5]. While being on call is not a new pattern of work, the human impacts of this working arrangement are under-represented in literature compared to other non-standard work patterns such as overtime, night, or rotating-shift rosters [2]. Despite the relative dearth of studies, there are consistent findings within the available literature particularly relating to the consequences for sleep, both in conjunction with and in the absence of actual calls and call-outs [5,6,7,8]. In addition to the impact on sleep, literature largely from the healthcare sector reports that on-call workers often experience stress, burnout, reduced job satisfaction [2,9,10,11], and a feeling of always being “on edge” [12], never knowing if or when they will be called. 

Given the differences in the division of unpaid, domestic labor between men and women [13], the unpredictable nature of on-call work appears especially difficult for women to navigate. Specifically, women spend nearly twice as long as men on childcare and domestic duties [13]. While few studies isolated the impacts of on-call work for men and women, emerging themes suggest domestic responsibility at home impacts on the experience of on-call work for women. For example, in the emergency services sector in Australia (where being on call is a necessary component of the work), family demands such as childcare responsibilities [14,15] and demands of emergency service roles conflicting with family demands [16] were reported. In addition, there are sex differences in causes of stress [17] and mental burnout [18] specifically in the on-call context. Collectively, the available research highlights being on call as a pattern of work which may pose uniquely different challenges for men and women. 

On-call work is fundamental to a number of industries, including healthcare and emergency response. Understanding the human impacts of on-call work is essential to inform decisions relating to the integration of on-call work into organizations and industries. One angle from which to approach this improved understanding is to identify factors that might reduce the likelihood of workers experiencing negative physical and psychological consequences associated with on-call work. In other non-standard patterns of work, for example, factors such as age, personality characteristics, and chronotype were associated with shift-work tolerance or coping [19,20]. In addition, there is evidence to suggest that preference for certain coping strategies—either engaged or disengaged—may predict how well individuals tolerate certain patterns of work in terms of the psychical and psychological outcome experience [21,22,23]. In a study in a cohort of nurses in Australia, it was found that those who had engaged coping styles were more likely to be in what was defined as the “healthy” (as opposed to less healthy) cluster of workers [22]. Importantly, unlike age, sex, or chronotype, coping style is modifiable; therefore, understanding current coping strategies more broadly will assist in the development of programs to support individuals and groups of workers in managing on-call work. 

Thus, this study aimed to explore how on-call work impacts workers’ domestic and social lives, and to determine the preferred coping style and strategies of these workers. We were particularly interested in whether the impacts of on-call work and how workers cope in the on-call context differed between males and females. 

## 2. Materials and Methods 

### 2.1. Survey Methodology

An anonymous online survey (Survey Monkey) was disseminated among industry connections, social media (Facebook, Twitter), and personal networks. While open to anyone identified as doing on-call work, targeted advertisement of the survey was aimed at the emergency services and health sectors, as on-call work is prevalent across these sectors [2,12]. Potential participants were provided with a survey link where they were first presented with an information sheet about the study. Individuals were invited to voluntarily participate, and were informed the survey would take approximately 15–20 min. Before beginning, participants completed a declaration confirming they were over eighteen years old, living in Australia, had read and understood the information provided, and consented to participate. Ethical approval was granted by CQ University’s Human Research Ethics Committee (H17/05-088). 

### 2.2. Participants

A total of 261 participants started the survey; however, two people did not do on-call work, so they were excluded. Thirty-one people started the survey but (a) did not answer any questions, or (b) did not answer any questions relevant for these analyses; thus, they were removed. This resulted in a final sample size of 228 participants.

### 2.3. Measures

#### 2.3.1. Demographic, Domestic, and Social Measures

Survey questions included basic demographics; the impact of on-call work for leisure, domestic, and non-domestic work, and people important to them, as well as how well workers believe they cope, and their preferred coping strategies. Questions regarding sleep are reported elsewhere [6]. Questions were largely closed-ended and forced choice, although a small number of open-ended questions were included, focusing on describing on-call commitment, and the factors which make it difficult for participants to cope with on-call work. Participant main employment and on-call employment roles were classified according to the Australian and New Zealand Standard Industrial Classification (ANZSIC) broad industry divisions and industry subdivisions to allow for future comparison across industries.

#### 2.3.2. Coping Strategies

The coping questions in the survey were adapted from those in the Standard Shiftwork Index [24] which establishes participants’ preferred coping style when managing problems or stressful situations. Respondents indicated their use of eight basic coping strategies in response to the question, “To what extent do you use the following strategies when you experience problems?”. The eight basic coping strategies were problem solving, cognitive restructuring, social support, expressing emotions, problem avoidance, wishful thinking, self-criticism, and social withdrawal when faced with problems. The scale included an item for each coping strategy with a five-point Likert scale for each item ranging from “not used” (score 1) through to “used a great deal” (score 5). The subscales were then used to sum scores for the engaged coping styles (problem solving, cognitive restructuring, social support, expressing emotions) and disengaged coping styles (problem avoidance, wishful thinking, self-criticism, and social withdrawal). This overall score then determined participants’ overall preferred coping style of “engaging” with their environment or “disengaging” to manage problems and/or stressful situations. The current study also incorporated a third category of “neutral” for those participants who displayed an equal preference to using engaged/disengaged coping strategies.

### 2.4. Data Analyses

The data were analyzed using IBM SPSS Statistics (Version 24.0; IBM Corp., 2016, New York, NY, USA). Due to the descriptive nature of the study, chi-square analyses were used to explore sex differences in demographic characteristics and self-reported domestic/social responsibilities, as well as each of the individual coping strategies and the overall coping style. 

## 3. Results

A total of 228 (57% male) workers who identified as working on call completed the survey. Participants were from a range of on-call professions, including emergency services (fire-fighters, state emergency service workers, paramedics), healthcare (doctors, nurses, midwives, allied health), the electrical sector, and information technology. The most prevalent industries of employment were healthcare and social assistance (*n* = 94, 41.2%), public administration and safety (*n* = 55, 24.1%), agriculture, forestry, and fishing (*n* = 17, 7.5%), and information, media, and telecommunications (*n* = 14, 6.1%). Socio-demographic and occupational characteristics of the sample by sex are shown in Table 1. 

A greater percentage of females (42.9% compared with 21.5%) indicated that they agreed/strongly agreed that they were solely responsible for running their household/caring for their family (χ^2^(2) = 12.3, *p* = 0.002) compared to the male respondents, who were more likely to work full-time. There were 46 respondents who did not have a partner (14.6% males and 27.6% females), and, for those that did, more than three-quarters (78.6%) of the sample indicated that their partner was (extremely/fairly) supportive of their on-call commitments. There were no differences in age, youngest person in the household, duration of on-call work, and whether their partner was supportive of on-call commitments between women and men (see Table 1). 

### 3.1. Perceived Impact of On-Call Work on Leisure, Domestic, Non-Domestic Duties, and Important People in Workers’ Lives

Workers reported that their on-call work interferes with aspects of their lives outside work (see Table 2). Perceived interference was higher in female respondents across all three aspects (leisure time use, χ^2^(2) = 12.41, *p* = 0.002; domestic activities in time off, χ^2^(2) = 8.77, *p* = 0.021; non-domestic activities in time off, χ^2^(2) = 12.75, *p* = 0.002). This was most pronounced in leisure time, with female respondents indicating that their on-call work interferes a lot/very much (57.1%) or somewhat (29.6%) with the sorts of things they would like to do in their leisure time. Domestic requirements in time off work were somewhat (41.8%) or a lot/very much (38.8%) impacted according to female respondents. In contrast, male respondents indicated that their leisure time was impacted somewhat (43.8%) or a lot/very much (33.8%) by on-call work, but over 50% felt that non-domestic requirements during time off were not at all/a bit impacted by their on-call work.

Respondents were also asked more broadly to indicate whether they perceived their on-call work negatively impacted on important people in their lives. Over half of respondents (male, 51.7%, *n* = 61; female, 55.6%, *n* = 50) irrespective of sex either agreed or strongly agreed that their on-call work negatively impacted on important people in their lives. On average, 15.9% (male, 17.8%, *n* = 21; female, 13.3%, *n* = 12) of participants disagreed/strongly disagreed. No sex differences were apparent in these responses (χ^2^(2) = 0.79, *p* = 0.674). 

### 3.2. Coping 

When asked the degree to which they agreed with the statement, “I cope well with on-call work”, responses were categorized into three groups (strongly agree/agree, neutral, disagree/strongly disagree). For men and women, the percentages who agreed/strongly agreed with the statement were 70.0% and 58.0%, respectively, with 15.8% of men and 14.2% of women indicating that they disagreed/strongly disagreed. There were no significant differences between the sexes for this question (χ^2^(2) = 3.92, *p* = 0.142).

The extent to which participants used the eight different coping approaches are presented below in Table 3. There were significant differences between males and females in terms of “working on solving problems in the situation” (χ^2^(1) = 0.04, *p* = 0.012), with more than three-quarters of the male respondents using the strategy “quite a bit/great deal” compared with just over half of females. In terms of “talking” as a strategy, differences between males and females were also significant, with 38.9% of females using the strategy “quite a bit/great deal” compared with 17.8% of males (χ^2^(1) = 0.04, *p* < 0.0001).

Depending on participant responses, points (1–5) were allocated for each coping strategy, with higher scores indicating a greater use of that strategy. Points for each of the four “engaged” (solve problems, reorganize way I look, emotions out, talk) and “disengaged” (avoid thinking, with situation would go away, criticize myself, spend time alone) coping styles were collated, giving each participant a total score for both engaged and disengaged coping. A participant’s preferred coping style (engaged or disengaged) was allocated if that style had the highest score. In the case where a participant had equal scores for both engaged and disengaged strategies, they were assigned as “neutral”. The majority (67.3%, *n* = 140) of the sample of workers were identified as having an engaged coping style (males, 67.8%, *n* = 80; females, 66.7%, *n* = 60), while 26.0% (*n* = 54) of the sample had a disengaged coping style (males, 26.3%, *n* = 31; females, 25.6%, *n* = 23). A neutral (neither engaged or disengaged) coping style was apparent in 7.8% of females (*n* = 7), and 5.9% of males (*n* = 7). There were no sex differences in preferred coping style (χ^2^(2) = 0.28, *p* = 0.870).

## 4. Discussion

The aims of this study were to establish a preliminary socio-demographic profile of the on-call workforce in Australia, to describe sex differences in term of the impact that on-call work has and how workers cope. In line with this current knowledge about unpaid, domestic labor in Australia [13], there were significant differences surrounding what level of responsibility participants had in terms of running their households. A greater proportion of female respondents (42.9%) indicated that they were solely responsible for running their household compared to 21.5% of males. This information is important for workers and employers managing on-call work; the unpredictable burden of on-call work may be more difficult to manage for individuals who are also managing their household—in this sample, this was largely the female respondents. 

Previous literature reported that the unpredictability of on-call work interferes with both work and non-work activities [12,15,16,17,25,26]. Qualitative studies in particular shed light on the ways that on-call workers limit or modify their activities, particularly leisure time, to ensure they can be contacted, and attend work if needed [12,15]. Findings from the current study are in line with previous research, with participants indicating that on-call work interferes with leisure, domestic, and non-domestic aspects of their lives (Table 2). Importantly, while both males and females feel the impact of on-call work, the degree of impact (“not at all” to “very much”) for each of the categories differed significantly between the sexes, with a greater proportion of females in each category reporting that the impact of on-call was felt “a lot/very much”. The discrepancy was largest in the leisure category, with 65% of males indicating that on-call impacted their leisure “not at all” or “a little” compared with 57% of females, who indicated that leisure was impacted “a lot” or “very much”. The sex differences regarding responsibility for running the household in the current sample, may shed light on why females feel a greater or at least differing impact of on-call work. As already discussed, a greater proportion of women in this cohort were more likely to be solely responsible for running their household, and, not surprisingly, may then find the impact on domestic activities greater than men. For leisure activities also, the added domestic burden for women, in combination with the unpredictable on-call load, possibly leaves less time and scope for what might be considered “non-essential” activities. 

The implications of the findings regarding leisure time, particularly for women, should be considered in the context of recovery and mental detachment from work. It is known that being unable to separate work from domestic life can impact the ability to relax and may contribute to poor well-being [27,28]. Additionally, being unable to engage in leisure activities (separate from work and domestic) may have a similar effect. Furthermore, the unpredictability and low control inherent to on-call work may not only disrupt leisure time, but also lead to poor work–life balance for workers [29], as work-time control is positively associated with work–life balance [30]. In turn, higher work–life conflict was found to be associated with increased fatigue and psychological symptoms [31]. While these were not explicitly explored in these analyses, the lack of control and how this infiltrates the time between calls is something that needs to be factored into on-call rostering. It was beyond the scope of this survey to gain detailed information about how participants spent their time, on or off call; thus, the need for more in-depth research exploring these potential differences in time use between men and women remains.

The survey also asked specifically about the impact of on-call work for people in the workers’ lives, and more than half the sample agreed or strongly agreed that on-call work had a negative impact on the important people in their lives. Again, this finding is supported by previous, limited literature, including our own qualitative studies [12,15,17,32] reporting negative impacts of on-call work on children and partners, particularly in terms of important events and time away from home. These data serve to reiterate what is known about on-call patterns of work in terms of the impact for the family unit. In addition to the negative effects for people close to the workers, these findings should be considered in terms of what it means for the support that family/friends can offer the worker, in both the short and longer term. Notably, in the current cohort, despite a majority reporting that important people were adversely impacted, nearly 80% of those with a partner (*n* = 182), indicated that their partners were supportive of their on-call commitments. Partner support may be one of a key set of characteristics or circumstantial factors that contribute to positive coping or experience with non-standard working arrangements such as on-call work. Indeed, support (from partners, children, friends, and workplace) was identified as a major theme associated with positive coping in a recent qualitative study of women on call in the emergency services [15].

An encouraging outcome from the current study was the finding that, when asked specifically whether they thought they coped well with on-call work, only 15% of both males and females indicated that they did not cope (disagree/strongly disagree with the statement). The majority of men and women either agreed/strongly agreed or were neutral toward the statement. These are important findings and may be explained in part by the types of coping styles and strategies used by the group. All respondents reported use of at least some engaged coping strategies specifically in relation to their on-call work. Importantly, when a total coping score was calculated, two-thirds of the sample (67%) was classified as having an overall “engaged” coping style. This is a positive outcome for on-call workers given the research that shows positive coping strategies are linked to enhanced well-being [33]. Furthermore, and more specifically, research demonstrated that individuals who adopt engaged or positive coping strategies are better placed to tolerate their non-standard working conditions [21,22]. It is possible that this observation is part of a type of healthy-worker effect [34], whereby those who are coping well and perhaps have suitable support remain in this type of work, which is also in line with our finding regarding partner support, where more than three-quarters of those with a partner indicated that they had their support in this role. This may enable organizations to identify possible training which may be adopted to help workers cope positively with on-call work, or assist to buffer the negative health impacts.

While both male and female respondents were equally “engaged” copers, there were differences in terms of preference for specific engaged coping strategies. For both men and women, problem solving was used most frequently, with a significantly greater proportion of males adopting that strategy compared to females. There were also significant sex differences when it came to use of talking about emotions, with a significantly greater proportion of women adopting this strategy. Identification of individuals’ preferred coping style is a valuable starting point for more tailored support or even intervention strategies; however, if we can ascertain that there are predictable sex differences in terms of preferred styles, targeted support strategies can begin even while we are extending knowledge in this context. 

### Limitations

While this research is the first to describe the consequences of on-call work in terms of impact and coping in Australia’s on-call workers, the study design does come with some limitations that should be taken into account when interpreting the results. The cross-sectional nature of the survey means that the data captured are just a “snapshot” of on-call work and its impacts, and are likely influenced by participants’ most recent experiences, which may or may not be representative of their overall experience. Longitudinal data can not only remove this bias, but also capture any changes to dependent variables over time. This should be a consideration for future research. We also acknowledge that forced-choice type questions, as many in this survey were, remove some depth to individuals’ experiences, and assume that responses will be the same in all circumstances. This is the first study to describe the Australian on-call workforce (including both volunteer and salaried workers), and, while the sample did represent many of the known sectors that utilize on-call patterns of work [2,5,12], it is not possible to determine whether the study sample was a representative group. Response bias may, therefore, be a factor for consideration when extrapolating these findings. Furthermore, the small sample size meant analyses were contained to descriptive statistics, and these should be viewed with the limited sample in mind. Larger-scale studies comprising greater numbers of on-call workers will facilitate further exploration of predictors of specific coping strategies, which will extend the application of these findings. 

Despite these limitations, this study provides an early perspective of the factors contributing to how well Australian on-call workers cope with their work. To improve on this initial description, further research should look to collect data from larger and/or more targeted samples, so meaningful statistical comparisons between, for example, different sectors or those with differing family responsibilities can be made. Future research may also consider the human impacts of on-call work and the factors which assist workers to positively cope (or buffer the effects) with these impacts, particularly regarding sex differences.

## 5. Conclusions 

This study reports basic demographic characteristics of Australian on-call workers; importantly, it is the one of only a few studies to specifically explore differences between men and women who do on-call work across a broad range of occupations. The study provides valuable insight not only into the differing ways in which males and females experience on-call work, but also into coping style and preferred coping strategies. Findings suggest that females perceive the impact of on-call work to be greater on all fronts (domestic, non-domestic, and leisure) and we propose that these differences may be related to the imbalanced domestic burden that exists between males and females in Australia. With further research, this information can contribute firstly to the identification of individuals who are vulnerable to on-call work, and secondly to the development of appropriate support strategies, which will be crucial for both recruitment and retention of individuals, both male and female, into roles with an on-call component. 

## Figures and Tables

**Table 1 ijerph-16-00370-t001:** Socio-demographic and occupational characteristics of respondents, highlighting statistically significant differences between male and female respondents for each characteristic.

Socio-Deomgraphic and Occupational Characteristic	Total *n* = 228		Male *n* = 130 (57%)		Female *n* = 98 (43%)		*p*
	*n*	%	*n*	%	*n*	%	
**Individual and household characteristics**							
Age (years)							0.085
18–24	23	10.1	19	14.6	4	4.1	
25–34	48	21.1	25	19.2	23	23.5	
35–44	50	21.9	30	23.1	20	20.4	
45–54	68	29.8	34	26.2	34	34.7	
55+	39	17.1	22	16.9	17	17.3	
Youngest person in the house							0.782
<5 years	33	14.5	18	13.8	15	15.3	
6–12 years	24	10.5	16	12.3	8	8.2	
13–18 years	26	11.4	15	11.5	11	11.2	
>18 years	145	63.6	81	62.3	64	65.3	
Sole responsibility for household							**0.002**
Agree	70	30.7	28	21.5	42	42.9	
Neither agree nor disagree	50	21.9	34	26.2	16	16.3	
Disagree	108	47.4	68	52.3	40	40.8	
**Occupational characteristics**							
Current work arrangements							**0.001**
Full-time	172	75.4	106	81.5	66	67.5	
Part-time	33	14.5	9	6.9	24	24.5	
Casual	10	4.4	5	3.8	5	5.1	
Other	13	5.7	10	7.7	3	3.1	
On-call part of main occupation							**<0.001**
Yes	176	77.2	88	67.7	88	89.8	
No	52	22.8	42	32.3	10	10.2	
Duration of on-call work							0.243
<12 months	17	7.5	8	6.2	9	9.2	
1–10 years	123	53.9	66	50.8	57	58.2	
>10 years	88	38.6	56	43.1	32	32.7	
Partner supportive of on-call commitments *							0.234
Yes	143	78.6	84	75.7	59	83.1	
No	39	21.4	27	24.3	12	16.9	

* Those without a partner were excluded from calculation. Significant results (*p* < 0.05) in bold.

**Table 2 ijerph-16-00370-t002:** Perceived interference of on-call work with leisure, domestic, and non-domestic duties.

How Much Does On-Call Work Interfere with	Total		Male (*n* = 130, 57%)		Female (*n* = 98, 43%)		*p*
	*n*	%	*n*	%	*n*	%	
*The sort of things you would like to do in your leisure time (e.g., sport activities, hobbies)*							**0.002**
Not at all/a bit	42	18.4	29	22.3	13	13.3	
Somewhat	86	37.7	57	43.8	29	29.6	
A lot/very much	100	43.9	44	33.8	56	57.1	
*The domestic things you have to do in your time off work (e.g., domestic tasks, children)*							**0.012**
Not at all/a bit	67	29.4	48	36.9	19	19.4	
Somewhat	88	38.6	47	36.2	41	41.8	
A lot/very much	73	32.0	35	26.9	38	38.8	
*The non-domestic things you have to do in your time off work (e.g., doctor, library, bank)*							**0.002**
Not at all/a bit	93	40.8	66	50.8	27	27.6	
Somewhat	77	33.8	38	29.2	39	39.8	
A lot/very much	58	25.4	26	20.0	32	32.7	

**Table 3 ijerph-16-00370-t003:** Frequency of coping style strategy stratified by sex.

Coping Strategy	Total (*n* = 228)	Male (*n* = 130, 57%)	Female (*n* = 98, 43%)	*p*
*n*	%	*n*	%	*n*	%
***I work on solving the problems in the situation***							**0.012**
Not used	1	0.5	0	0.0	1	1.1	
Used a little/somewhat	67	32.2	29	24.6	38	42.2	
Used quite a bit/a great deal	140	67.3	89	75.4	51	56.7	
***I reorganize the way I look at the situation so things do not look so bad***							0.918
Not used	13	6.3	8	6.8	5	5.6	
Used a little/somewhat	114	54.8	65	55.1	49	54.4	
Used quite a bit/a great deal	81	38.9	45	38.1	36	40.0	
***I let my emotions out***							0.394
Not used	35	16.8	23	19.5	12	13.3	
Used a little/somewhat	135	64.9	76	64.4	59	65.6	
Used quite a bit/a great deal	38	18.3	19	16.1	19	21.1	
***I talk to someone about how I am feeling***							**<0.0001**
Not used	28	13.5	23	19.5	5	5.6	
Used a little/somewhat	124	59.6	74	62.7	50	55.6	
Used quite a bit/a great deal	56	26.9	21	17.8	35	38.9	
***I avoid thinking or doing anything about the situation***							0.880
Not used	68	32.7	40	33.9	28	31.1	
Used a little/somewhat	123	59.1	68	57.6	55	61.1	
Used quite a bit/a great deal	17	8.2	10	8.5	7	7.8	
***I wish the situation would go away or somehow be over with***							0.850
Not used	59	28.4	35	29.7	24	26.7	
Used a little/somewhat	111	53.4	61	51.7	50	55.6	
Used quite a bit/a great deal	38	18.3	22	18.6	16	17.8	
***I criticize myself for what is happening***							0.950
Not used	40	19.2	23	19.5	17	18.9	
Used a little/somewhat	108	51.9	62	52.5	46	51.1	
Used quite a bit/a great deal	60	28.8	33	28.0	27	30.0	
***I spend time alone***							0.858
Not used	36	17.3	19	16.1	17	18.9	
Used a little/somewhat	112	53.8	65	55.1	47	52.2	
Used quite a bit/a great deal	60	28.8	34	58.8	26	28.9

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
