# Peer review of "Understanding the Differing Impacts of On-Call Work for Males and Females: Results from an Online Survey"

_ijerph, 2019, doi:10.3390/ijerph16030370_

Round 1
Reviewer 1 Report
This paper provides significant information on how on-call work impacts different aspects of workers’ domestic, non-domestic and social lives and in particular, how males and females adopt different coping strategies to respond to the requirements of this job.
The paper demonstrated an adequate understanding of the existing literature in this field and expresses clearly its cases and uses appropriate technical language and clear structure.
However, the paper is merely descriptive of the results at this point and doesn’t attempt to make any comparisons between the different sub-groups. I believe such comparisons would make the discussion more interesting. Therefore, I think the following suggestions would improve the clarity and importance of the manuscript on these sections:
It would be interesting for the readers to see a comparison between those males and females who do have sole domestic responsibility. The paper reports the rate for males and females and that a higher number of females are solely responsible for running the household/care for the family, but no further comparisons between those males and females who have this role are reported. It would be interesting to see if the coping styles for those males and those females are similar.
Moreover, it would also be interesting to see comparisons between those males and females who do not have a partner to support them, or between those males/females working full-time vs. those working part-time, and how they cope with being on-call. I understand that the sample sizes would be smaller for these comparisons but the current results may be skewed by these differences and, where possible, would be interesting to see if similar results for gender differences would be obtained. If some of these comparisons are not all possible, please explain why.
In relation to reporting/presentation of the results, the reporting of the chi-square significance values needs more clarification in the tables. It’s not clear in the tables which comparison the significance values reported correspond to; in other words, the comparison between males and females on which dimension (agree /not sure/disagree etc.) did it yield significance? Could you please report this more clearly either on the tables or in each relevant section in the text?
Finally, the results for the question whether on-call work has an impact on important people in their lives (p.5, lines 149-154) are not included in the table above (Table 2), is there a reason for this? It would be good to include these results either in Table 2 or in any table the authors feel more appropriate.
The limitations and discussion section are well written and appropriate but would be enriched with the suggested comparisons.
Author Response
We thank the reviewer for their considered and constructive comments. We have addressed each point int he attached document.

Reviewer 2 Report
This is an interesting piece of work for which the authors are to be congratulated.
It covers an important area of workplace stress, notably for First Responders.
The authors should strongly consider the possibility of a longitudinal study in this area, although a lot of work might be needed in getting a good N of participants.
It was unfortunate this study did not include a measure of fatigue such as the OFER scale which could indicate Persistent Fatigue tendency, which underpins maladaptive chronic health problems.
Overall a very publishable piece of work
Author Response
We thank the reviewer for their review of our manuscript and for their positive comments regarding its interest and importance to the field.
Reviewer 3 Report
Interesting paper, original subject, although a limited number of respondents (228) but interesting sectors. it would be good to pursue the research in specific sectors and maybe increase respondents to be able to compare these sectors. emergency services (fire-fighters,
120 state emergency services workers, paramedics), health care (doctors, nurses, midwives, allied health),
121 electrical sector and information technology. The most prevalent industries of employment were
122 health care and social assistance (n=94, 41.2%), public administration and safety (n=55, 24.1%),
123 agriculture, forestry and fishing (n=17, 7.5%) and information, media and telecommunications (n=14,
124 6.1%). Sociodemographic
Author Response
We thank the reviewer for their review of out manuscript and acknowledgment of its originality and interest to readership.
